# Multi-petahertz electron interference in Cr:Al$_2$O$_3$ solid-state material

Hiroki Mashiko [1], Yuta Chisuga[1,2], Ikufumi Katayama[2], Katsuya Oguri[1], Hiroyuki Masuda[1,2], Jun Takeda [2] & Hideki Gotoh[1]

Lightwave-field-induced ultrafast electric dipole oscillation is promising for realizing petahertz ($10^{15}$ Hz: PHz) signal processing in the future. In building the ultrahigh-clock-rate logic operation system, one of the major challenges will be petahertz electron manipulation accompanied with multiple frequencies. Here we study multi-petahertz interference with electronic dipole oscillations in alumina with chromium dopant (Cr:Al$_2$O$_3$). An intense near-infrared lightwave-field induces multiple electric inter-band polarizations, which are characterized by Fourier transform extreme ultraviolet attosecond spectroscopy. The interference results from the superposition state of periodic dipole oscillations of 667 to 383 attosecond (frequency of 1.5 to 2.6 PHz) measured by direct time-dependent spectroscopy and consists of various modulations on attosecond time scale through individual electron dephasing times of the Cr donor-like and Al$_2$O$_3$ conduction band states. The results indicate the possible manipulation of petahertz interference signal with multiple dipole oscillations using material band engineering and such a control will contribute to the study of ultrahigh-speed signal operation.

[1] NTT Basic Research Laboratories, 3-1 Morinosato Wakamiya, Atsugi, Kanagawa 243-0198, Japan. [2] Department of Physics, Graduate School of Engineering, Yokohama National University, 79-5 Tokiwadai, Hodogaya, Yokohama 240-8501, Japan. Correspondence and requests for materials should be addressed to H.M. (email: mashiko.hiroki@lab.ntt.co.jp)

To date, high-speed signal processing is performed by electronic devices using semiconductor-based field-effect transistors driven by radio-frequency (RF) electric fields[1]. The speed of such electronic devices has reached a limit at the terahertz ($10^{12}$ Hz: THz) regime because of the response time of band energy modulation with RF electric fields[2]. Lightwave-field control opens up a new opportunity to speed up the frequency into the petahertz ($10^{15}$ Hz: PHz) regime because the ultrafast electric dipole variation with inter-band polarization activates an electronic device with instantaneous optical switching from an insulator to conductor or vice versa[3].

A powerful way to monitor the temporally evolving coherent electronic motion in solid-state materials is to use modern material-sensing technology with an isolated attosecond pulse (IAP)[4–7]. In a previous study, we observed the electric dipole oscillation with a single petahertz frequency component using gallium-nitride (GaN) semiconductor[6]. To build ultrahigh-clock-rate logic operation systems, petahertz electron manipulation accompanied with multiple frequencies is the next challenge. Our approach for the signal manipulation is to use the interference provided by the multiple electron motions through material band engineering.

Here we study a petahertz interference constructed with near-infrared (NIR) lightwave-field-induced multiple electronic dipole oscillations in alumina with chromium dopant (Cr:Al$_2$O$_3$) and reveal by Fourier transform extreme ultraviolet attosecond spectroscopy (FTXUV) combined with an IAP.

## Results

**Experimental condition and properties of Cr:Al$_2$O$_3$.** Trigonal (rhombohedral) α-Al$_2$O$_3$ is a typical electric insulator with a wide-bandgap[8]. It is commonly used in the manufacture of semiconductor epitaxial wafers, owing to its hardness, high thermal conductivity, and resistance to optical damage. In this experiment, the α-Al$_2$O$_3$ is doped with the Cr material during the single-crystalline α-Al$_2$O$_3$ crystal growth. The Cr$^{3+}$ ions produce a donor-like intermediate level for the Al$_2$O$_3$ host material[9,10]. Figure 1a shows a schematic of the experimental setup for the FTXUV based on the transient absorption spectroscopy (for details, see the Methods section and Supplementary Note 1). The collinearly propagated IAP (44-eV center photon energy with 192-as duration[11]) and NIR pulse (1.55-eV center photon energy with 7-fs duration) are focused onto the target (for the IAP characterization, see the Supplementary Fig. 1). The timing jitter between IAP and NIR pulse is 23 as at the root mean square over 12 h in this pump-probe system[6]. The thin 36-nm-thick target without a substrate is manufactured from 400-μm-thick bulk target by mechanical polishing and ion beam milling. The electric fields of the IAP and NIR pulse are injected perpendicular to c-axis of the Al$_2$O$_3$. Figure 1b shows the measured absorption coefficient α in the target (red filled-circles and solid line) at room temperature. For comparison, that of a high-purity Al$_2$O$_3$ sample with the bandgap energy $E_g$ of 8.7 eV[8] is shown by a red dashed line. The Urbach tail[12] region corresponding to the donor-like state is 5 to 8.7 eV. The measured absorption trace agrees with previous reports on Cr:Al$_2$O$_3$ solids[10,13]. The Al$_2$O$_3$ host material

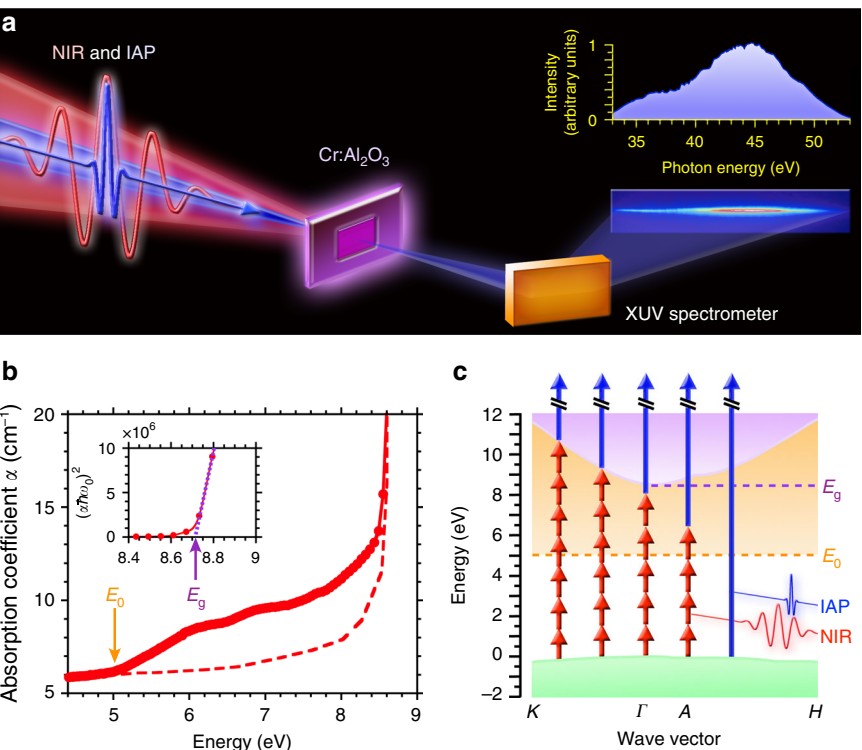

**Fig. 1** Experimental setup and properties of Cr:Al$_2$O$_3$. **a** Schematic view of experimental setup for the Fourier transform extreme ultraviolet attosecond spectroscopy (FTXUV) based on transient absorption spectroscopy. An IAP with 192-as duration (44-eV center photon energy) and a NIR pulse with 7-fs duration (1.55-eV center photon energy) were used. The target is 36-nm-thick bulk single-crystalline α-Al$_2$O$_3$ with Cr dopant. The transmitted IAP from the target was detected by an extreme ultraviolet (XUV) spectrometer. **b** Measured absorption coefficient α in the target (red filled circles and solid line). For comparison, the red dashed line shows that of the high-purity Al$_2$O$_3$ sample from ref. [8]. The inset shows $(\alpha\hbar\omega_0)^2$ as a function of energy. The $\hbar$ and $\omega_0$ are the Dirac constant and the angular frequency of the injected light source in the absorption spectroscopy. The Urbach energy $E_0$ of the initial donor-like state and bandgap energy $E_g$ of the Al$_2$O$_3$ host material are 5 eV (orange arrow) and 8.7 eV (purple arrow), respectively. **c** Energy level diagram of the target. The green shaded area is the VB of the Al$_2$O$_3$ at K, Γ, A, and H points in wave vector[14]. The orange shaded area corresponds to the Cr donor-like state. The purple shaded area is the CB state of Al$_2$O$_3$ host material. Blue and red arrows show the IAP and NIR pulse, respectively

has the atomic number density of $2 \times 10^{22}$ cm$^{-3}$. The density of Cr dopant is $2 \times 10^{17}$ cm$^{-3}$, which was measured by the secondary ion mass spectrometry (SIMS) (see the Supplementary Fig. 2). The estimated doping level is approximately $1 \times 10^{-3}$ at. % (10 ppm). Figure 1c shows the energy level diagram of the Al$_2$O$_3$[14]. Here the intensity of the NIR pulse is $2 \times 10^{12}$ W/cm$^2$ on the target, which induces multiple inter-band polarizations from the valence band (VB) state to the Cr donor-like state and the Al$_2$O$_3$ conduction band (CB) state. Simultaneously, the IAP with high photon energy allows the excitation of electrons from the VB, donor-like, and CB states. The electrons are finally excited to the high-level CB states in the Al$_2$O$_3$, which have hyperfine states and behave as quasi-continuum states.

**Multi-petahertz interference in Cr:Al$_2$O$_3$.** Figure 2a shows the measured multi-petahertz interferogram in the Cr:Al$_2$O$_3$. The trace shows a deviation of optical density ($\Delta$OD) with and without the NIR pulse as a function of temporal delay (for the definition of $\Delta$OD, see Methods section). The trace exhibits the characteristic temporal modulation in the whole energy region. The coherence of the superposition state created by electrons from the VB, donor-like, and CB states leads to quantum interference, which results in a temporal modulation of the IAP absorption spectrum in the whole photon energy region[6]. To confirm the electron transition process, we analyze the Keldysh parameter[15] $\gamma$. The lightwave-field-induced electron tunneling is defined by the laser intensity, which corresponds to $\gamma \ll 1$. In this experiment, the $\gamma$ estimated from the minimum transition level of the Urbach energy ($E_0 = 5$ eV) is 4.7, which is $\gamma \gg 1$. Thus, the multiphoton process dominates the inter-band polarization, and the use of wide-bandgap materials makes it possible to induce the multiple petahertz oscillations via the multiphoton process[6]. In principle, such temporal modulation as shown in Fig. 2a is capable of producing the ultrafast conductivity variation in solid-state material[3]. Figure 2b shows the integrated line profiles for the photon energy regions of 38–46 eV in Fig. 2a. The profile has a

variety of separations on an attosecond time scale, which are produced by the interference built on the superposition state of multiple electric dipole oscillations with different periodicities.

Figure 2c shows the energy components with Fourier transformation of the temporal delay axis in Fig. 2b. The dotted lines correspond to the $4\hbar\omega$, $5\hbar\omega$, $6\hbar\omega$, and $7\hbar\omega$ photon energy components of the NIR pulse ($\hbar\omega = 1.55$ eV). The multiphoton process with the NIR pulse produces these specific energy components in the donor-like (orange shaded area) and the CB (purple shaded area) states. Here the $4\hbar\omega$ component has higher signal intensity than the $5\hbar\omega$ one because it is the first resonance state for a perturbative multiphoton process. However, the higher-order $6\hbar\omega$ and $7\hbar\omega$ components recover the signal intensity because the absorption coefficient drastically increases above the bandgap energy of Al$_2$O$_3$ ($E_g = 8.7$ eV). In principle, the absorption coefficient in the intermediate levels can be controlled with a variety of dopant materials[9] and their doping level[13]. Consequently, this result strongly indicates that the interfered petahertz signal is manipulatable with the photon energy tuning of the driving laser and desirable with material band engineering in solid-state material.

**Multiple dipole oscillations and electron dephasing.** Next, we analyze temporal components in each electron transition. Figure 3a shows the absorption spectra in Fig. 2a after the windowed Fourier transform. The center window energies of the Fourier filtering are 6.2 ($4\hbar\omega$), 7.7 ($5\hbar\omega$), 9.2 ($6\hbar\omega$), and 10.8 eV ($7\hbar\omega$). The window bandwidth is applied with $\pm0.35$ eV with the NIR bandwidth taken into account. Figure 3b shows each integrated line profile for the photon energy regions of 38–46 eV in Fig. 3a. The oscillation periodicities are 667 ($4\hbar\omega$), 537 ($5\hbar\omega$), 450 ($6\hbar\omega$), and 383 as ($7\hbar\omega$). The converted driving frequencies correspond to 1.5, 1.9, 2.2, and 2.6 PHz, respectively. The electron dynamics of 383-as periodicity has the shortest electric dipole oscillation ever recorded in a direct time-dependent measurement based on a pump-and-probe scheme[16]. The result also indicates that the

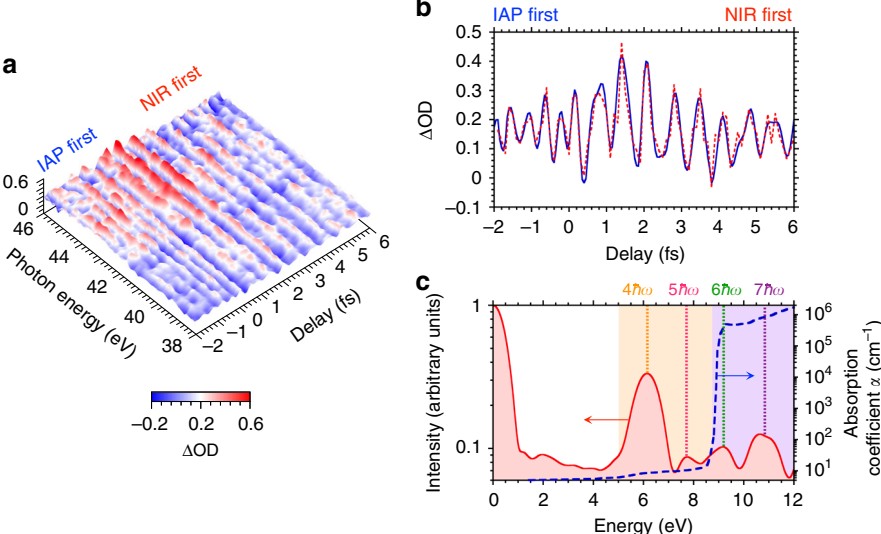

**Fig. 2** Multi-petahertz interferogram and energy components in Cr:Al$_2$O$_3$. **a** Measured interferogram based on transient absorption spectroscopy. The trace shows the deviation of optical density ($\Delta$OD) with and without the NIR pulse as a function of the temporal delay. The trace is averaged over ten measurements. **b** Graph showing the integrated line profile (red dashed line) for the 38–46 eV photon energy region in **a**. The blue solid line corresponds to the interpolation process line profile. **c** Line profile spectrum (red solid line and shaded area) with Fourier transformation for the temporal delay axis in **b**. The dotted lines correspond to the photon energies of 6.2 ($4\hbar\omega$), 7.7 ($5\hbar\omega$), 9.2 ($6\hbar\omega$), and 10.8 eV ($7\hbar\omega$). The $\omega$ is the center angular frequency of the NIR pulse. The $\hbar\omega$ corresponds to 1.55 eV. The blue dashed line shows the absorption coefficient $\alpha$ from our measurement (<8.7 eV) and ref. [33] (>8.7 eV). The orange and purple shaded areas correspond to the donor-like and CB states, respectively

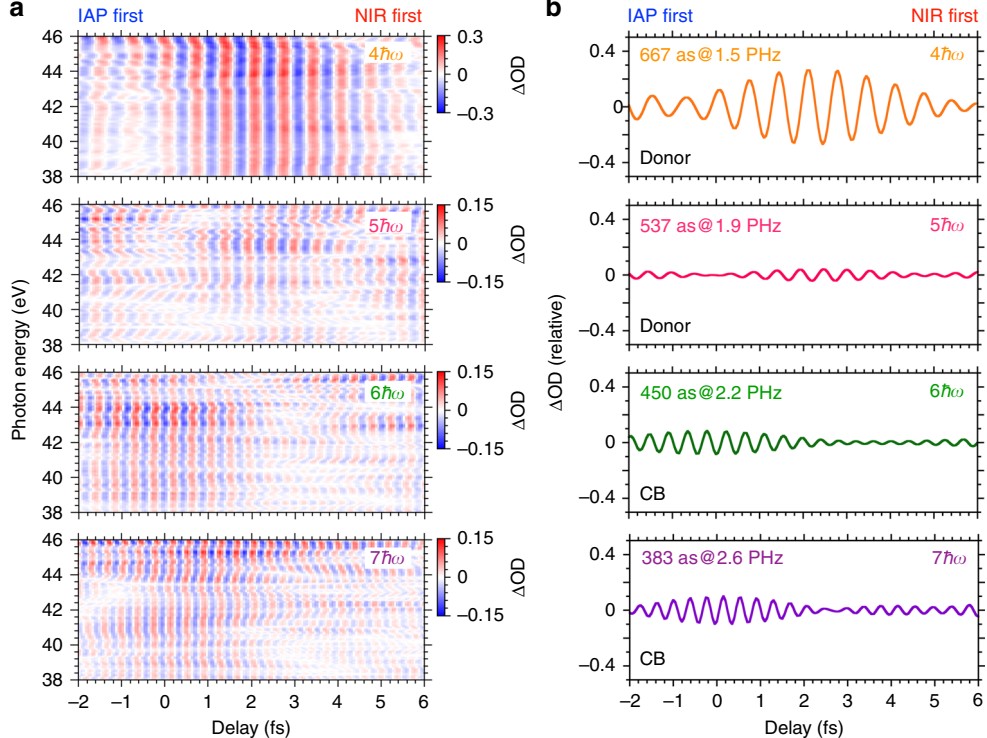

**Fig. 3** Temporal structure of each energy component. **a** Temporal structures in Fig. 2a after windowed Fourier transform. The window energies of Fourier filtering are 6.2 ($4\hbar\omega$), 7.7 ($5\hbar\omega$), 9.2 ($6\hbar\omega$), and 10.8 eV ($7\hbar\omega$). The applied window energy width is ±0.35 eV, taking into account the NIR bandwidth. The $4\hbar\omega$ and $5\hbar\omega$ correspond to the donor-like state. The $6\hbar\omega$ and $7\hbar\omega$ correspond to the CB state. **b** Traces showing the integrated line profile for the photon energy regions of 38–46 eV in **a**. The electric dipole oscillations with periodicities of 667 ($4\hbar\omega$), 537 ($5\hbar\omega$), 450 ($6\hbar\omega$), and 383 as ($7\hbar\omega$) correspond to frequencies of 1.5, 1.9, 2.2, and 2.6 PHz, respectively

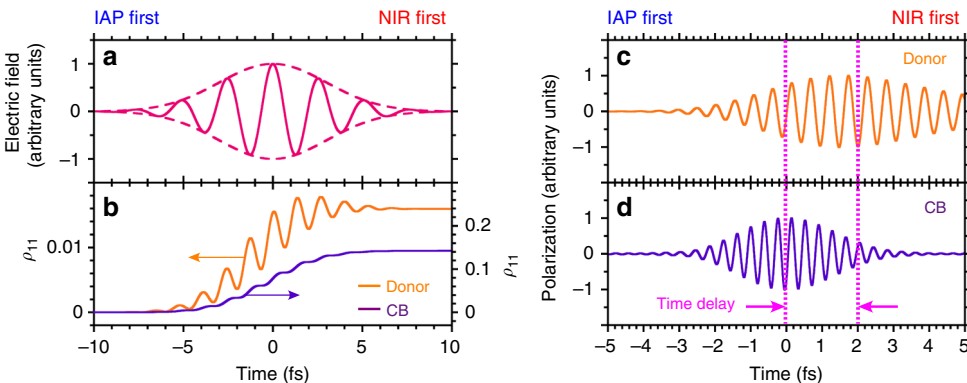

**Fig. 4** Calculated time-dependent density matrix element and polarization. **a** The applied electric field waveform of the pulse with 7-fs duration. **b** Populations of the excited state ($\rho_{11}$) in the donor-like ($5\hbar\omega$) and CB ($7\hbar\omega$) resonances with parameter values of $\mu F/\hbar\omega = 0.59$, $\tau_{CB} = 0.2$ fs and $\tau_{Donor} = 3$ fs. Fourier filtered polarizations in **c** donor-like and **d** CB resonances. The pink dotted lines and arrows show the time delay between the donor-like and CB components

donor-like components ($4\hbar\omega$ and $5\hbar\omega$) have a relative time delay of approximately 2 fs from the CB components ($6\hbar\omega$ and $7\hbar\omega$). The relative time delay (2 fs) and the different oscillation periodicities (667–383 as) produce various temporal modulations on the interferogram, as shown in Fig. 2b.

For a better understanding of this time delay of approximately 2 fs, we numerically simulated the resonant high-order polarizations using the model in a two-level system[17,18], which consists of the ground state and the excited state either resonant to the donor-like state ($5\hbar\omega$) or resonant to the CB state ($7\hbar\omega$) (for details, see Supplementary Note 4). The calculated results with

the parameter value of $\mu F/\hbar\omega = 0.59$, where $\mu$ and $F$ correspond to the strengths of the transition dipole moment and the applied NIR driving laser field (7-fs duration), are shown in Figs. 4a, b. Here, the parameter value is consistent with that estimated from the Rabi frequency under our experimental condition. We found that the time delay could be tuned via the polarization dephasing times $\tau$ parameterized in the donor-like and CB states. Assuming that the CB state ($\tau_{CB} = 0.2$ fs) has the faster dephasing time of 2.8 fs relative to the donor-like state ($\tau_{donor} = 3$ fs), the observed time delay of approximately 2 fs is reproduced well, as shown in Figs. 4c, d. The value of the dephasing time at room temperature

is comparable to those obtained in previous pump-probe experiments using silicon (Si)[5] and gallium arsenide (GaAs)[19] solid-state materials, where the ultrafast dephasing time of 5–14 fs observed originates from intra-band electron–electron scattering. Note that when the dephasing times in the donor-like and CB states are equivalent, the buildup times of the polarization are almost identical in this simulation even though the states have different transition energies. Generally, the dephasing time in the spatially localized energy state is much longer than that in the band energy state[20,21]. Thus, the spatially localized Cr donor-like state with the low doping level has many fewer relaxation channels in the unoccupied state compared with the CB state. Consequently, the delayed buildup of the polarization could be observed in the resonant polarization of the spatially localized Cr donor-like state. In addition, the dephasing time is commonly explained by the density- and energy-dependent damping rate[22], which is proportional to cube root of the excited carrier density $n$; the value of $n$ depends on the fluence of pump pulse and the absorption coefficient of the target[23,24]. The linear absorption coefficient $\alpha$ in the CB state drastically increases higher than the fourth order of magnitude compared with the donor-like state, as shown in Fig. 2c. Thus, the excited carrier density in the CB state might be higher than the donor-like state even though the nonlinear multiphoton excitation. The different carrier densities could produce individual dephasing times in the donor-like and CB states. In any case, the important issue is the tuneable time delay induced by polarization behavior, which can be designed by band-structure control through the kind of dopant material and doping level. This tuneability will further assist with the flexibility of the petahertz signal manipulation.

## Discussion

Meanwhile, the intra-band polarization is another possible source of the inducement of ultrafast dipole oscillation. The intra-band polarization could already be contained in the measured dynamics, which makes it difficult to discuss intra-band and inter-band electric motions separately[25]. Here, since the measured dynamics already has high-order nonlinear processes (4–7$\hbar\omega$ photon energies), the ponderomotive energy[26] $U_\mathrm{P}$ might be a useful parameter for qualitatively discussing the intra-band motion because it has been well used in terms of high-order harmonic generation (HHG) from solid-state material[27–30]. The $U_\mathrm{P}$ is proportional to the intensity and square of the wavelength of the driving laser. High harmonics are typically generated by a low-intensity (approximately $1 \times 10^{12}$ W/cm$^2$), long-wavelength (middle-infrared and far-infrared) driving laser[27–29] or by a high-intensity (approximately $1 \times 10^{13}$ W/cm$^2$), short-wavelength NIR one[30]. In this experiment, the estimated $U_\mathrm{P}$ is 0.12 eV, which is extremely low compared to the $U_\mathrm{P} = 1$ to 23 eV in HHG experiments[27–30]. Therefore, in the measured dynamics, the inter-band polarization could have a large contribution compared with the intra-band polarization. Note that, the concept of the ponderomotive energy $U_\mathrm{P}$ contains the effective mass approximation corresponding to the parabolic band in the HHG theory.

Previously, the petahertz electric dipole oscillations have been discussed in terms of the electric-field reconstruction of high-order harmonics emitted from polycrystalline quartz (SiO$_2$)[30] in solid-state materials. In contrast, the present study with the FTXUV method provides the direct time-domain observation of electronic dipole oscillations with nonlinear polarization. This property will provide sensitive detection and ultrafast manipulation. Since the dipole oscillation is the origin of light-matter interaction, the benefits are related to the control of various optical phenomena through the dielectric polarization. In addition, the directly identified time dependence in the solid reveals

electric dipole oscillations with transition energy of up to 10.8 eV (7$\hbar\omega$) in this experiment. The energy region covers the bandgap energies for almost all semiconductor and insulator materials[31]. Therefore, this study lays the essential groundwork for exploring the band states in solid-state material, and the controllable time dependence resulting from the material band engineering will be important for developing petahertz digital electronics in the future.

## Methods

**Definition of deviation of optical density (ΔOD).** In the FTXUV material sensing, the transient absorption spectrum at temporal delay $t$ between the IAP and NIR pulse is given by: $\Delta\mathrm{OD}(\omega_\mathrm{IAP},t) = \log[I_\mathrm{NIRout}(\omega_\mathrm{IAP},t)/I_\mathrm{NIRin}(\omega_\mathrm{IAP},t)]$, where $\omega_\mathrm{IAP}$ is the laser frequency of the IAP, $I_\mathrm{NIRout}(\omega_\mathrm{IAP},t)$ is the absorption spectrum of the IAP without the NIR pulse, and $I_\mathrm{NIRin}(\omega_\mathrm{IAP},t)$ is that with the NIR pulse added. Consequently, the $\Delta\mathrm{OD}(\omega_\mathrm{IAP},t)$ monitors the absorbance deviation by the NIR pulse.

**FTXUV experimental setup.** In this experiment, a few-cycle pulse (1.55-eV center photon energy with 7-fs duration) from a Ti:sapphire laser was used for high-harmonic generation and as a pump-NIR pulse for the FTXUV (see Supplementary Note 1). The probe-IAP (44-eV center photon energy with 192-as duration) is generated by the double optical gating (DOG) technique[32] using argon atom (see Supplementary Note 2). In this pump-probe system, the timing jitter is 23 as at the root mean square over 12 h, which is monitored by a helium neon laser[6]. The IAP spectrum transmitted from the Cr:Al$_2$O$_3$ target is detected by a regular extreme ultraviolet spectrometer (see Supplementary Note 3). The spectrometer resolution is 120 meV at 45.5-eV photon energy[11].

**Data availability**. The data that support the findings of this study are available from the corresponding authors.

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

## Acknowledgements

We thank T. Tawara and A. Tanaka of NTT Basic Research Laboratories and T. Ikeda of NTT Advance Technology Inc. for $Cr:Al_2O_3$ sample preparation and the material investigation. This work was supported by JSPS KAKENHI Grant No. 16H05987 and 16H02120.

## Author contributions

Y.C., H. Masuda, and H. Mashiko performed the experiments. I.K. performed the calculation. H. Mashiko, K.O., J.T., and H.G. planned and coordinated the project. H. Mashiko and Y.C. wrote the manuscript with contributions from all authors.

## Additional information

**Competing interests:** The authors declare no competing interests.

