## [Peer Review File · Nature Communications]

Reviewer #1 (Remarks to the Author):

Review on “Multi-petahertz electron interference in Cr: Al_2O_3 solid-state material” by Mashiko et al.
Using Fourier transform XUV attosecond spectroscopy, H. Mashiko *et al.* demonstrate the fastest dipole oscillations (1.5-2.6 PHz) ever directly measured in the time domain. To achieve these record speeds, the authors used material band engineering, which is a crucial novel step for the further development of petahertz signal processing.

Material band engineering is achieved by doping Al_2O_3 with Cr ($\sim 1 \times 10^{-5}$ weight %), which acts as donor. The dopant material and density allows to control the absorption coefficient in the donor and acceptor states. This work thus indicates that the combination of photon energy tuning of the driving laser together with band engineering (via doping) allows to manipulate petahertz signal processing in a controlled fashion.

The paper is nicely written, the results are valid and very important for both science and technology. The authors demonstrate manipulation of petahertz signals by electronic band engineering, for the first time to my knowledge. This work is therefore timely and important for the further development of petahertz digital electronics. Overall, I strongly recommend publication of this work in Nature Communications. However, the authors should address the following points in the published version of the manuscript:

(1) extremely short dephasing times, excitation-induced dephasing (page 3 and Supplement, page 2):
In their theoretical simulations, Mashiko *et al.* used extremely short phenomenological dephasing times: $\tau = 3$ fs for donor states and $\tau = 0.2$ fs for th

The authors attribute the different dephasing times to differences in electron-electron scattering in the donor and CB states, which is probably correct. The authors argue “In previous experiments using silicon¹¹ and gallium arsenide¹² solid-state material, the intraband electron-electron scattering produced the ultrafast polarization dephasing time of several femtoseconds (5-14 fs) at room temperature. The electronic state with higher energy generally has the faster dephasing time¹³. In addition, the faster dephasing in Al_2O_3 may be because of the additional dephasing due to the existence of the donor state. The higher density of states in the CB state may also accelerate the dephasing.”

Here, in their simulations and discussion the authors completely ignore the carrier-density (n_c) dependence of the dephasing time (‘excitation-induced dephasing’). The average time between electron-electron collisions is proportional to $n_c^{-1/3}$, and for high carrier densities n_c it can go down to few femtoseconds in gallium arsenide [Q.T. Vu *et al.*, PRL **92**, 217403 (2004)]. The intensities (3×10^{12} W/cm²) used in that PRL were comparable to the NIR intensity of 2×10^{12} W/cm² on target in the present manuscript. Thus, when comparing to experiments on GaAs [25], the authors should also comment on the possibility of excitation-induced dephasing in GaAs for intensities of few 10^{12} W/cm².

(2) range of validity of ponderomotive energy (page 3):

Note that in solids, the concept of the ponderomotive energy is only valid, as long as the effective mass approximation (corresponding to a parabolic band) is valid, i.e., only close to the band center. This should be made more clear in the manuscript to avoid confusion of the readers.

(3) minor typos:

page 2, line 11 from bottom: “acceptor” instead of
“accepter” page 3, line 6: “ponderomotive” instead of
“poderomotive”

Supplement, page 2, line 3 from bottom: “dephasing” instead of “dephaseing”

Reviewer #2 (Remarks to the Author):

In the manuscript “Multi-Phz electron interference in Cr:alpha-Al₂O₃ solid-state material,” H. Mashiko et al perform attosecond streaking on a thin chromium doped sapphire sample. This experiment builds on their previous work (ref 5), where they investigated GaN. This paper is well written and is interesting to the attosecond and condensed matter communities, and I think it should be published in Nature Communications.

By performing attosecond streaking on chromium doped sapphire, the authors are able to measure a change in the optical density, as measured in the modulation of the XUV spectral amplitude, as a function of NIR delay. By having the XUV first or the NIR first, Mashiko et al claim that the modulation period changes and this is a condition of the states which play a role. The data is nice, and their analysis is thorough.

However, I have a problem with their discussion in the change in the modulation period. On page 2, discussing Fig 2, the authors state “This profile obviously contains multiple electric dipole oscillations with different periodicities...” I do not find the change obvious, and this change in modulation is, I feel, the main point of the paper. As shown in Fig 2b, the authors have chosen two points, a peak and a valley, and show that they change separation. I would say that to make this statement “obvious,” then this modulation should be discussed in a Gabor transform (windowed Fourier transform) to show the change in the modulation period. I feel that by selecting different peaks/troughs, I could find nearly any separation I would like between 200 – 500 as, and I feel that the authors have chosen the largest difference, but not a representative difference. I think that their conclusions can be much stronger if this change in OD modulation is more carefully presented. Furthermore, the 500 as peak/valley separation implies a period of 1 fs, which is just under the 3rd harmonic of the fundamental, and yet this harmonic is not present in Fig 2c. I feel like this justifies my concern of claiming that this change is “obvious.”

My other question is the statement made in the main text and then repeated in the Supplemental Information that: “The electronic state with higher energy generally has the faster dephasing time.” This is an interesting statement, but I cannot find the justification of this statement in the cited reference (ref 26). I feel like this statement requires further justification to be made, either by a more suitable reference or calculations. The dephasing time in condensed matter attoscience is of interest to the community, and this statement should be made with more care.

Typo: second to last line in page 2: “the multiphoton process dominates the interband polarization and it combined with...” “it” should be either “when” or “if”.

Reviewer #3 (Remarks to the Author):

The manuscript entitled "Multi-petahertz electron interference in Cr:alpha-Al₂O₃ solid-state material" (Ref. No.: NCOMMS-17-26398-T) reports results of an interesting pump-probe experiment with near infrared (NIR) and isolated attosecond pulses (IAP). The target is a doped wide bandgap solid state material, where the dopant levels are between the conduction and valence band edges.

The NIR and the IAP pulses – although have wide spectra – correspond to (central) photon energies that are considerably different and consequently interact with the material in a different manner. Several NIR photons are needed to bridge the bandgap, while the energy of the photons in the IAP is a few times larger than the gap. The action of the different pulses is demonstrated to depend on the time delay between them, indicating quantum mechanical coherence effects on a very short time scale (on the order of PHz in frequency domain). Additionally, the secondary radiation emitted by the sample could be related to the "source transition", i.e., either to valence band (VB)-conduction band (CB) or dopant level-CB transitions, with a measurable time delay between the two signals. This effect is particularly interesting, and I am impressed that a relatively simple, two-level open quantum system model could recover this phenomenon.

In conclusion, the manuscript reports interesting results obtained using an impressively stable laser source. Besides state of the art experimental techniques, the authors also apply a transparent model to verify their statements. Based on these, I think that the paper can be published in Nature Communications, provided the authors address the questions/comments below.

Questions:

1. Considering the experimental results, it would be nice to elaborate how the time delay between signals originating from different transitions were seen to arrive with a delay to the detector. Was this result based simply on the central frequency of the signal, or the authors used additional considerations?
2. In the theoretical model, as far as I could see, besides the dephasing times, the energy difference between the two levels was also different for the VB-CB and VB-dopant level transitions. Did this energy difference have any observable effect on the dynamics, or the time delay between the signals originating from different transitions could have been assigned solely to the difference of the dephasing times?

Remarks (optional):

3. I am not completely convinced that the connection between the experimental results and the possibility of PHz electronics is as direct as expressed in the text. Either this should be elaborated by a more detailed explanation, or less emphasized. In my opinion, the experimental results are important enough to be published without referring to these (far future?) applications.
4. In view of Phys. Rev. B 96, 035112 (2017), the notions of inter- and intraband parts of the dynamics should be considered with a special care. Based on the gage dependence of these quantities, the authors may consider putting less emphasis on the distinction between inter- and intraband contributions.

Manuscript: NCOMMS-17-26398-T

Title: Multi-petahertz electron interference in Cr:Al₂O₃ solid-state material

Author: H. Mashiko, Y. Chisuga, I. Katayama, K. Oguri, H. Masuda, J. Takeda, and H. Gotoh

Reply to all referees:

First, we thank all the referees for their fruitful comments and evaluations regarding our manuscript. Following the referees' comments and suggestions, we have significantly revised our manuscript as follows:

- (1) We have added new Figure S2 and the corresponding explanations in supplementary section S3. The figure shows the measured atomic number density of Cr dopant using secondary ion mass spectrometry (SIMS).
- (2) We have replaced the word "dopant concentration" with the generic science term "doping level" throughout the manuscript.
- (3) We have changed the unit from "wt. % (weight %)" to "at. % (atomic %)" and "ppm".
- (4) Since we do not deeply discuss the material structure dependence (e.g. α -, β -, and γ -), we have removed the " α -" from Al₂O₃ except in the material explanation in the updated manuscript.
- (5) Since the Cr dopant does not produce electrons or holes in the conduction band in this work, we have replaced "donor state" with "donor-like state" or "donor-like intermediate level" to avoid misleading the reader.

As shown below, we have addressed on a point-by point basis the individual issues raised by the referees.

Reviewer #1:

Using Fourier transform XUV attosecond spectroscopy, H. Mashiko et al. demonstrate the fastest dipole oscillations (1.5-2.6 PHz) ever directly measured in the time domain. To achieve these record speeds, the authors used material band engineering, which is a crucial novel step for the further development of petahertz signal processing.

Material band engineering is achieved by doping α -Al₂O₃ with Cr ($\sim 1 \times 10^{-5}$ weight %), which acts as donor. The dopant material and density allows to control the absorption coefficient in the donor and acceptor states. This work thus indicates that the combination of photon energy tuning of the driving laser together with band engineering (via doping) allows to manipulate petahertz signal processing in a controlled fashion.

The paper is nicely written, the results are valid and very important for both science and technology. The authors demonstrate manipulation of petahertz signals by electronic band engineering, for the first time to my knowledge. This work is therefore timely and important for the further development of petahertz digital electronics. Overall, I strongly recommend publication of this work in Nature Communications. However, the authors should address the following points in the published version of the manuscript:

REPLY: We thank the referee for his/her high evaluation. In the revised manuscript, we have tried our best to make the suggested revisions.

(1) extremely short dephasing times, excitation-induced dephasing (page 3 and Supplement, page 2):

In their theoretical simulations, Mashiko et al. used extremely short phenomenological dephasing times: $\tau=3$ fs for donor states and $\tau=0.2$ fs for the conduction band (CB) state.

The authors attribute the different dephasing times to differences in electron-electron scattering in the donor and CB states, which is probably correct. The authors argue "In previous experiments using silicon¹¹ and gallium arsenide¹² solid-state material, the intraband electron-electron scattering produced the ultrafast polarization dephasing time of several femtoseconds (5-14 fs) at room temperature. The electronic state with higher energy generally has the faster dephasing time¹³. In addition, the faster dephasing in α -Al₂O₃ may be because of the additional dephasing due to the existence of the donor state. The higher density of states in the CB state may also accelerate the dephasing."

Here, in their simulations and discussion the authors completely ignore the carrier-density (n_c) dependence of the dephasing time ('excitation-induced dephasing'). The average time between electron-electron collisions is proportional to $n_c^{-1/3}$, and for high carrier densities n_c it can go down to few femtoseconds in gallium arsenide [Q.T. Vu et al., PRL 92, 217403 (2004)]. The intensities (3×10^{12} W/cm²) used in that PRL were comparable to the NIR

intensity of 2×10^{12} W/cm² on target in the present manuscript. Thus, when comparing to experiments on GaAs [25], the authors should also comment on the possibility of excitation-induced dephasing in GaAs for intensities of few 10^{12} W/cm².

REPLY: We thank the referee for the suggestions and fully agree with the referee. The discussion of excitation-induced dephasing is very important. Following the referee's suggestion, we have added the following sentences on page 3, line 38 in the revised manuscript: "In addition, the dephasing time is commonly explained by the density- and energy-dependent damping rate²², which is proportional to cube root of the excited carrier density n ; the value of n depends on the fluence of pump pulse and the absorption coefficient of the target^{23,24}. The linear absorption coefficient α in the CB state drastically increases higher than the forth order of magnitude compared with the donor-like state, as shown in Fig. 2(c). Thus, the excited carrier density in the CB state might be higher than the donor-like state even though the nonlinear multiphoton excitation. The different carrier densities could produce individual dephasing times in the donor-like and CB states." In addition, we have added the suggested reference as ref. (22) [Q.T. Vu et al., PRL 92, 217403 (2004)]. Furthermore, we have added a discussion in supplementary section S4 on page 3, lines 6-20.

(2) range of validity of ponderomotive energy (page 3):

Note that in solids, the concept of the ponderomotive energy is only valid, as long as the effective mass approximation (corresponding to a parabolic band) is valid, i.e., only close to the band center. This should be made more clear in the manuscript to avoid confusion of the readers.

REPLY: We agree with the referee. We have added an explanation on page 4, line 1 in the manuscript (including response to reviewer #3): "Meanwhile, the intra-band polarization is another possible source of the inducement of ultrafast dipole oscillation. The intra-band polarization could already be contained in the measured dynamics, which makes it hard to separately discuss intra- and inter-band electric motions²⁵. Here, since the measured dynamics already has high-order nonlinear processes ($4-7\hbar\omega$ photon energies), the ponderomotive energy²⁶ U_p might be a useful parameter for qualitatively discussing the intra-band motion because it has been well used in terms of high-order harmonic generation (HHG) from solid-state material²⁷⁻³⁰. The U_p is proportional to the intensity and square of the wavelength of the driving laser. High harmonics are typically generated by a low-intensity (approximately 1×10^{12} W/cm²), long-wavelength (middle- and far-infrared) driving laser²⁷⁻²⁹ or by a high-intensity (approximately 1×10^{13} W/cm²), short-wavelength (near-infrared) one³⁰. In this experiment, the estimated U_p is 0.12 eV, which is extremely low compared to the $U_p = 1-23$ eV in HHG experiments²⁷⁻³⁰. Therefore, in the measured dynamics, the inter-band polarization could have a large contribution compared with the intra-band polarization. Note that, the concept of the ponderomotive energy U_p contains the effective mass approximation corresponding to the parabolic band in the HHG theory."

(3) minor typos:

page 2, line 11 from bottom: "acceptor" instead of "accepter"

page 3, line 6: “ponderomotive” instead of “poderomotive”

Supplement, page 2, line 3 from bottom: “dephasing” instead of “dephaseing”

REPLY: We apologize for our English misspellings. We have corrected those misspelled words in the revised manuscript.

REPLY: Again, we thank the referee for all of his/her comments and considerations for our manuscript.

Reviewer #2:

In the manuscript “Multi-Phz electron interference in Cr: α -Al₂O₃ solid-state material,” H. Mashiko et al perform attosecond streaking on a thin chromium doped sapphire sample. This experiment builds on their previous work (ref 5), where they investigated GaN. This paper is well written and is interesting to the attosecond and condensed matter communities, and I think it should be published in Nature Communications.

By performing attosecond streaking on chromium doped sapphire, the authors are able to measure a change in the optical density, as measured in the modulation of the XUV spectral amplitude, as a function of NIR delay. By having the XUV first or the NIR first, Mashiko et al claim that the modulation period changes and this is a condition of the states which play a role. The data is nice, and their analysis is thorough.

REPLY: We first thank the referee for the appreciative comments and important suggestions. We have tried to make the suggested revisions.

However, I have a problem with their discussion in the change in the modulation period. On page 2, discussing Fig 2, the authors state “This profile obviously contains multiple electric dipole oscillations with different periodicities...” I do not find the change obvious, and this change in modulation is, I feel, the main point of the paper. As shown in Fig 2b, the authors have chosen two points, a peak and a valley, and show that they change separation. I would say that to make this statement “obvious,” then this modulation should be discussed in a Gabor transform (windowed Fourier transform) to show the change in the modulation period. I feel that by selecting different peaks/troughs, I could find nearly any separation I would like between 200 – 500 as, and I feel that the authors have chosen the largest difference, but not a representative difference. I think that their conclusions can be much stronger if this change in OD modulation is more carefully presented.

We thank the referee for the important comments. We totally agree with you. The temporal separations on the interferogram [Fig. 2(b)] should be directly connected to the discussion of the temporal components ($4-7\hbar\omega$) after the windowed Fourier transform [Fig. 3] and to the discussion of the dephasing time [Fig. 4]. To clearly describe our claims, we have moved the location of the discussion of Fig. 2 to Fig. 4. Thus, we first discuss the Keldysh parameter on page 2, lines 29–36. We have also added an explanation on page 2, line 36 in the revised manuscript: “Figure 2(b) shows the integrated line profiles for the photon energy regions of 38-46 eV in Fig. 2(a). The profile has a variety of separations on an attosecond time scale, which are produced by the interference built on the superposition state of multiple electric dipole oscillations with different periodicities.” and on page 3, line 12 in the manuscript: “The result also indicates that the donor-like components ($4\hbar\omega$ and $5\hbar\omega$) have a relative time delay of approximately 2 fs from the CB components ($6\hbar\omega$ and $7\hbar\omega$).

The relative time delay (2 fs) and the different oscillation periodicities (667-383 as) produce various temporal modulations on the interferogram, as shown in Fig. 2(b)."

Furthermore, the 500 as peak/valley separation implies a period of 1 fs, which is just under the 3rd harmonic of the fundamental, and yet this harmonic is not present in Fig 2c. I feel like this justifies my concern of claiming that this change is "obvious."

REPLY: To avoid misleading the reader, we have removed the discussion of 200-500 as and arrows from Fig. 2(b). In this part of the manuscript, we have emphasized the modulation change from multiple oscillations with different periodicities.

My other question is the statement made in the main text and then repeated in the Supplemental Information that: "The electronic state with higher energy generally has the faster dephasing time." This is an interesting statement, but I cannot find the justification of this statement in the cited reference (ref 26). I feel like this statement requires further justification to be made, either by a more suitable reference or calculations. The dephasing time in condensed matter attoscience is of interest to the community, and this statement should be made with more care.

REPLY: We are very sorry but we mislabeled the ref. number; the number should be ref. 27 [Levenson M. D., and Kano, S. S., Introduction to nonlinear laser spectroscopy (Academic Press, New York, 1988)] instead of ref. 26 in the previous version of manuscript. This textbook explains well the dephasing time T_2 (or τ) treatment in the density matrix. The decoherence effect induced by lower states is briefly explained, however, the issue of "The electronic state with higher energy generally has the faster dephasing time", has not been deeply treated. To avoid the unclear discussion, we have removed the sentence and the reference.

Typo: second to last line in page 2: "the multiphoton process dominates the interband polarization and it combined with..." "it" should be either "when" or "if".

REPLY: Following the referee's suggestions, we have corrected the sentence on page 2, line 32 in the revised manuscript: "Thus, the multiphoton process dominates the inter-band polarization, and the use of wide-bandgap materials makes it possible to induce the multiple petahertz oscillations via the multiphoton process⁴."

REPLY: We believe that all the concerns raised are now satisfactorily resolved.

Reviewer #3:

The manuscript entitled “Multi-petahertz electron interference in Cr:alpha-Al₂O₃ solid-state material” (Ref. No.: NCOMMS-17-26398-T) reports results of an interesting pump-probe experiment with near infrared (NIR) and isolated attosecond pulses (IAP). The target is a doped wide bandgap solid state material, where the dopant levels are between the conduction and valence band edges.

The NIR and the IAP pulses – although have wide spectra – correspond to (central) photon energies that are considerably different and consequently interact with the material in a different manner. Several NIR photons are needed to bridge the bandgap, while the energy of the photons in the IAP is a few times larger than the gap. The action of the different pulses is demonstrated to depend on the time delay between them, indicating quantum mechanical coherence effects on a very short time scale (on the order of PHz in frequency domain). Additionally, the secondary radiation emitted by the sample could be related to the “source transition”, i.e., either to valence band (VB)-conduction band (CB) or dopant level-CB transitions, with a measurable time delay between the two signals. This effect is particularly interesting, and I am impressed that a relatively simple, two-level open quantum system model could recover this phenomenon.

In conclusion, the manuscript reports interesting results obtained using an impressively stable laser source. Besides state of the art experimental techniques, the authors also apply a transparent model to verify their statements. Based on these, I think that the paper can be published in Nature Communications, provided the authors address the questions/comments below.

REPLY: We first thank the referee for the appreciative comments regarding our submitted manuscript.

Questions: 1. Considering the experimental results, it would be nice to elaborate how the time delay between signals originating from different transitions were seen to arrive with a delay to the detector.

REPLY: Although we may not fully understand this question, we have tried our best to response to it.

In our experiment, we could only determine the relative time delay (~2 fs) between signals from **VB-donor** and **VB-CB** transitions. However, we stress that this relative time delay is the key for the observed multi-petahertz polarizations that are the central issues in our paper.

In the case of the absolute time delay between **input IAP** and **VB-donor/VB-CB** transitions, we have to simultaneously monitor the IAP and NIR fields, e.g., using the IAP-NIR cross-correlation based on attosecond streak (photoelectron detection). As a result, we may evaluate the absolute time zero between **IAP-NIR** on the delay axis. However, there are a lot of parameters to be determined coming from both the optical absorption and photoelectron detections: the former contains the group and phase

delays depending on sample thickness and refractive index, and the latter yields the quantum delay between transitions and the electron propagation delay in the sample interface. In addition, the simultaneous detection with high accuracy is technically very difficult.

Was this result based simply on the central frequency of the signal, or the authors used additional considerations?

REPLY: We simply monitor the transmitted IAP spectrum at around center energies (38-46 eV). We did not apply any additional consideration in the data process, e.g., we added neither an offset of bandgap energies nor an offset of the NIR photon energy.

2. In the theoretical model, as far as I could see, besides the dephasing times, the energy difference between the two levels was also different for the VB-CB and VB-dopant level transitions. Did this energy difference have any observable effect on the dynamics, or the time delay between the signals originating from different transitions could have been assigned solely to the difference of the dephasing times?

REPLY: In this simulation, when the dephasing times in the donor-like and CB states are equivalent, the buildup times of the polarization are almost identical even through their transition energies are different. Here, if the dephasing time is shorter than the pulse duration of the NIR, the polarization almost follows the harmonics of the NIR electric field. On the contrary, if the dephasing time is much longer than the pulse duration, the resonant polarization builds up until the NIR passes by. Therefore, the peak position of the time-domain polarization will be delayed if the dephasing time is long compared with the pulse duration. This is exactly what we observed in the experiment, where the low-order harmonics (4th and 5th) are delayed by ~2 fs compared with the high-order harmonics (6th and 7th).

We have added the discussion on page 3, line 31 in the revised manuscript: “Note that when the dephasing times in the donor-like and CB states are equivalent, the buildup times of the polarization are almost identical in this simulation even though the states have different transition energies.”. Also, we have revised a discussion in supplementary section S4 on page 2, lines 41-47.

Remarks (optional): 3. I am not completely convinced that the connection between the experimental results and the possibility of PHz electronics is as direct as expressed in the text. Either this should be elaborated by a more detailed explanation, or less emphasized. In my opinion, the experimental results are important enough to be published without referring to these (far future?) applications.

REPLY: To improve the speed of data processing, electronics technologies will be gradually replaced with optical technologies such as optical telecommunications, optical computing, and so on. Although the concept of an optical transistor driven by

the electric light-wave field is an important issue in the recent work, our findings will contribute to far-distant future applications in ultrafast optoelectronics, as the referee suggested. Therefore, following the referee's suggestion, we have revised the abstract with less stress: "Lightwave-field-induced ultrafast electric dipole oscillation is promising for realizing petahertz (10^{15} Hz: PHz) signal processing in future¹⁻⁵. In building the ultrahigh-clock-rate logic operation system, one of the major challenges will be petahertz electron manipulation accompanied with multiple frequencies." and "The interference constructed with the superposition state of periodic dipole oscillations of 667-381 as (frequencies of 1.5-2.6 PHz), the fastest ever measured in direct time-dependent spectroscopy, produces a variety of modulations on an attosecond time scale through individual electron dephasing times of the Cr donor-like and Al_2O_3 conduction band states. The results strongly indicate that the petahertz signal is manipulatable with interference built on multiple dipole oscillations and desirable with material band engineering, which will contribute to the study of ultrahigh-speed signal operation.".

4. In view of Phys. Rev. B 96, 035112 (2017), the notions of inter- and intraband parts of the dynamics should be considered with a special care. Based on the gage dependence of these quantities, the authors may consider putting less emphasis on the distinction between inter- and intraband contributions.

REPLY: We agree with the referee. The dynamics could contain the intra-band motion. Thus, we have changed the explanations on page 4, line 1 in the revised manuscript (including the response to reviewer #1) and added ref. 25 [Phys. Rev. B 96, 035112 (2017)]: "Meanwhile, the intra-band polarization is another possible source of the inducement of ultrafast dipole oscillation. The intra-band polarization could already be contained in the measured dynamics, which makes it hard to separately discuss intra- and inter-band electric motions²⁵. Here, since the measured dynamics already has high-order nonlinear processes ($4-7\hbar\omega$ photon energies), the ponderomotive energy²⁶ U_p might be a useful parameter for qualitatively discussing the intra-band motion because it has been well used in terms of high-order harmonic generation (HHG) from solid-state material²⁷⁻³⁰. The U_p is proportional to the intensity and square of the wavelength of the driving laser. High harmonics are typically generated by a low-intensity (approximately 1×10^{12} W/cm²), long-wavelength (middle- and far-infrared) driving laser²⁷⁻²⁹ or by a high-intensity (approximately 1×10^{13} W/cm²), short-wavelength (near-infrared) one³⁰. In this experiment, the estimated U_p is 0.12 eV, which is extremely low compared to the $U_p = 1-23$ eV in HHG experiments²⁷⁻³⁰. Therefore, in the measured dynamics, the inter-band polarization could have a large contribution compared with the intra-band polarization. Note that, the concept of the ponderomotive energy U_p contains the effective mass approximation corresponding to the parabolic band in the HHG theory.".

REPLY: We believe that all the concerns raised are now satisfactorily resolved.

REPLY: Finally, we thank the referees again for their input. Following the reviewers' comments, we were able to significantly improve our manuscript. We believe that all the concerns raised by the referees are now resolved.

REVIEWERS' COMMENTS:

Reviewer #1 (Remarks to the Author):

The authors have carefully addressed all points raised in my previous report in their response and adequately revised the manuscript. I can therefore recommend publication in Nature Communications.

On page 3, line 42, it should be "fourth" instead of "forth"

Reviewer #2 (Remarks to the Author):

In the revised manuscript "Multi-petahertz electron interference in Cr:Al₂O₃ solid-state material," H. Mashiko et al address the scientific issues raised by this referee. I feel that the modifications made to the main text of the manuscript make this suitable to be published in Nature Communications.

However, I do feel that that some of the changes made to the abstract need to be modified. For example, in the first sentence there is a missing "the" before "future." I also find the last sentence difficult to parse.

With these small modifications I recommend for this manuscript to be published in Nature Communications.

Reviewer #3 (Remarks to the Author):

Having read the new version of the manuscript entitled "Multi-petahertz electron interference in Cr:alpha-Al₂O₃ solid-state material" (Ref. No.: NCOMMS-17-26398-T), I think that the authors carefully addressed my concerns. In their response, the authors answered all my questions, and I find the answers detailed and satisfactory. The manuscript was also modified appropriately. Based on this, and the merits of the paper that were detailed in my previous report, I suggest publishing it in Nature Communications as it is.

Manuscript: NCOMMS-17-26398-A

Title: Multi-petahertz electron interference in Cr:Al₂O₃ solid-state material

Author: H. Mashiko, Y. Chisuga, I. Katayama, K. Oguri, H. Masuda, J. Takeda, and H. Gotoh

Reply to all referees:

We thank all the referees again for their fruitful comments and evaluations regarding our manuscript. As shown below, we have minor revised on a point-by point basis the individual issues raised by the referees.

Reviewer #1:

The authors have carefully addressed all points raised in my previous report in their response and adequately revised the manuscript. I can therefore recommend publication in Nature Communications.

REPLY: Again, we thank the referee for all of his/her considerations.

On page 3, line 42, it should be "fourth" instead of "forth"

REPLY: We apologize again for our English misspellings. We have corrected the misspelled word in the revised manuscript.

Reviewer #2:

In the revised manuscript "Multi-petahertz electron interference in Cr:Al₂O₃ solid-state material," H. Mashiko et al address the scientific issues raised by this referee. I feel that the modifications made to the main text of the manuscript make this suitable to be published in Nature Communications.

REPLY: We thank the referee for the appreciative comments and suggestions.

However, I do feel that that some of the changes made to the abstract need to be modified. For example, in the first sentence there is a missing "the" before "future." I also find the last sentence difficult to parse.

REPLY: Following the referee's suggestions, we have added "the" and corrected the sentence on abstract in the revised manuscript: "The results indicate that the petahertz interference signal is manipulatable with multiple dipole oscillations via material band engineering, which will contribute to the study of ultrahigh-speed signal operation."

With these small modifications I recommend for this manuscript to be published in Nature Communications.

REPLY: We believe that all the concerns are satisfactorily resolved.

Reviewer #3:

Having read the new version of the manuscript entitled "Multi-petahertz electron interference in Cr:Al₂O₃ solid-state material" (Ref. No.:

NCOMMS-17-26398-T), I think that the authors carefully addressed my concerns. In their response, the authors answered all my questions, and I find the answers detailed and satisfactory. The manuscript was also modified appropriately. Based on this, and the merits of the paper that were detailed in my previous report, I suggest publishing it in Nature Communications as it is.

REPLY: We thank the referee again for the appreciative comments regarding our submitted manuscript.